# Changes in Body Mass Index and Their Associations with Psychological Distress, Worries, and Emotional Eating during the COVID-19 Pandemic: A Norwegian Cohort Study

**DOI:** 10.3390/nu15173834

**Published:** 2023-09-02

**Authors:** Elaheh Javadi Arjmand, Jens Christoffer Skogen, Jørn Henrik Vold, Silvia Eiken Alpers, Erik Kristoffer Arnesen, Silje Mæland, Lars Thore Fadnes

**Affiliations:** 1Department of Global Public Health and Primary Care, Faculty of Medicine, University of Bergen, 5020 Bergen, Norway; jorn.henrik.vold@helse-bergen.no (J.H.V.); silje.maeland@uib.no (S.M.); lars.fadnes@uib.no (L.T.F.); 2Bergen Addiction Research, Department of Addiction Medicine, Haukeland University Hospital, 5021 Bergen, Norway; silvia.eiken.alpers@helse-bergen.no; 3Department of Health Promotion, Norwegian Institute of Public Health, 5808 Bergen, Norway; jens.christoffer.skogen@fhi.no; 4Centre for Evaluation of Public Health Measures, Norwegian Institute of Public Health, 0473 Oslo, Norway; 5Alcohol and Drug Research Western Norway, Stavanger University Hospital, 4068 Stavanger, Norway; 6Division of Psychiatry, Haukeland University Hospital, 5021 Bergen, Norway; 7Department of Nutrition, Institute of Basic Medical Sciences, Faculty of Medicine, University of Oslo, 0317 Oslo, Norway; e.k.arnesen@medisin.uio.no

**Keywords:** body mass index, emotional eating, psychological distress, physical activity, COVID-19 pandemic

## Abstract

In 2020, the COVID-19 pandemic highlighted obesity’s long-term rise. Some of the impacts of the pandemic were increased psychological distress, emotional eating, higher consumption of high-sugar foods and drinks, and a more sedentary lifestyle. This study examined BMI changes over time and their associations with psychological distress and lifestyle changes. This population-based cohort study had 24,968 baseline participants and 15,904 and 9442 one- and two-year follow-ups, respectively. Weight, height, psychological distress, high-sugar foods and drinks, physical activity, and emotional eating were assessed. These factors and BMI were examined at baseline and over time. We used mediation analyses and structural equation modeling to determine how psychological distress affects BMI. The mean BMI was 25.7 kg/m^2^ at baseline and 26.2 kg/m^2^ at two years. High psychological distress, daily emotional eating, and low physical activity were associated with higher BMI at baseline and higher yearly increases in BMI compared to reference levels. Emotional eating mediated 33% of the psychological distress BMI effect. Overall, BMI increased during the pandemic. Psychological distress during the pandemic was linked to weight gain partly through emotional eating. This association remained strong over time during different stages of the pandemic.

## 1. Introduction

In the past four decades, the prevalence of overweight and obesity has grown dramatically [1]. Obesity is linked to the higher occurrence of several chronic illnesses, which entails an increase in morbidity and a major reduction in life expectancy [2,3,4]. In 2020, the increasing challenges of overweight and obesity coincided with the COVID-19 pandemic [5].

The COVID-19 pandemic disrupted daily life due to the disease itself and the preventive measures taken by the authorities to control the infection (lockdown, social distancing, home office, etc.) [6,7]. Although these measures were necessary to prevent the spread of the virus, as reported by studies in the early phases of the pandemic, they have also been reported to have had a psychological impact, such as contributing to increased levels of anxiety, depressive symptoms, perceived stress, post-traumatic stress symptoms, and, more generally, psychological distress [8,9,10,11]. Psychological distress can be viewed as an umbrella term that covers multiple common psychological conditions (such as those mentioned above) [12], and it has been documented that it co-occurs with obesity [13,14,15]. Several possible processes may contribute to obesity in a chronic stressful situation, such as decreased physical activity, changes in stress-related hormones, decreased length and quality of sleep, and changes in eating behaviors towards more unhealthy food choices and food cravings [16,17]. A multi-cohort study in Britain evaluated the long-term impact of the COVID-19 pandemic on psychological distress [18]. The authors reported that the pre-existing trend of higher psychological distress during mid-life (30 to 45 years old) and lower distress towards older age was interrupted by the COVID-19 pandemic [19]. Also, they showed an increase in psychological distress was highest in females [18].

Individuals who restrain their eating to achieve weight loss often struggle to control their food intake under psychological stress [20,21]. Their reliance on cognitive control rather than physiological cues makes them vulnerable to uncontrolled eating [22]. Hence, stressful situations, such as the recent pandemic, may increase the likelihood of weight gain for these individuals [23]. A study on weight change during self-quarantine showed that risk factors for weight gain during self-quarantine were snacking after dinner, lack of dietary restraint, and eating in response to stress [24]. In another study in the US, the longitudinal weight change in adults during the pandemic was assessed, and there was a mean increase in weight of 0.62 kg over 6 months between April and October 2020. Those who gained weight generally reported higher levels of anxiety, less control over their cravings, and a higher intake of snack food items during the lockdown period and after [25].

Increases in body mass index (BMI) are linked to high stress, as well as emotional eating [26,27]. The term “emotional eating” describes eating habits that are brought on by feelings other than hunger [28], such as feeling angry, depressed, or bored [29]. A study on eating habits during lockdown in Italy showed that the impact of isolation on emotional well-being is linked to emotional eating and, relatedly, higher BMI [30]. People who struggle with emotional eating might suppress their feelings by eating, and in this situation, they frequently prefer energy-dense, high-sugar, and high-fat food items that induce feelings of pleasure [31,32,33]. Long-term consumption of these types of foods could result in obesity and eventually related health risks, like diabetes and cardiovascular disease [34].

Studies during the early phases of the pandemic have shown a concerning number of adults reporting weight gain in different countries [8,24]. In this study, we assess the longitudinal change in BMI and its associations with emotional eating, intake of high-sugar food and drinks, and physical activity. Based on our previous work, we assumed that emotional eating and intake of high-sugar food and drinks would be associated with psychological distress and worries [35]. We, therefore, created a hypothesis model in which high psychological distress can increase BMI through changes in emotional eating and physical activity and unhealthy food choices (Figure 1).

The objective of this study was to assess BMI changes during different phases of the COVID-19 pandemic and investigate their associations with age, sex, psychological distress, worries, emotional eating, high-sugar food and drink intake, and physical activity.

## 2. Methods

### 2.1. Participants and Data Collection

The data were collected as a part of the Bergen in Change cohort study (BiE study). The impact of the COVID-19 pandemic and the non-pharmaceutical measures implemented were studied by surveying 81,170 individuals out of a total adult population of 224,000 in Bergen, Western Norway. Regarding age and sex distribution, the sample accurately reflected the population at large. A total of 29,535 people (response rate = 36%) agreed to take part in the study at the initial time point in April 2020 (t_0_), and of those, 84% (t_0_, *n*_0_ = 24,968) filled out the entire questionnaire, including weight and height (while those questionnaires missing these data were excluded). Further data were gathered in January 2021 (t_1_, *n* = 15,904) and May 2022 (t_2_, *n* = 9442). Several COVID-19-related preventive measures were put in place at t_0_, four to six weeks following the first wave of the pandemic in Norway. These included regulations on quarantine and social isolation; the closure of schools, museums, and gyms; and the requirement that people work from home. At the second time point (t_1_), there was some loosening in these restrictions, while at the final time point (t_2_), almost all the restrictions and pandemic precautions previously put in place were ceased.

### 2.2. Measures

Using the Web-based platform SurveyXact, the questionnaire was distributed to the invited participants by email and short text messages (SMS). The relevant items on the questionnaire that were used for this paper were: sex, age, weight, height, educational level, economic status, worries related to COVID-19, psychological distress, emotional eating, and intake of high-sugar food and drinks, which are described in Appendix A. In summary, worries related to health were measured by asking participants about worries related to them, their family, or relatives becoming infected by COVID-19. Financial worries were measured by the participants’ level of worry about losing their jobs or seeing a decline in personal finances. The level of worries was dichotomized (0 = no or some worries or 1 = substantial worry). We used the Hopkins Symptom Checklist (SCL-10) to evaluate psychological distress over the previous week, with a mean SCL-10 score of 1.85 representing clinically significant psychological distress [36]. For the measurement of unhealthy food choices, respondents were asked to consider their typical levels of consumption of high-sugar drinks and foods over the preceding month. Cakes, biscuits, sweets, and candy were given as examples of high-sugar foods in the questionnaire, whereas soft drinks and soda were given as examples of high-sugar drinks. High-sugar food and drink intake ranged from 0 (rare/never in the last month) to 1 (daily) in this model. Additionally, emotional eating (EE) was measured by having participants recall how often, over the previous seven days, they had engaged in comfort eating or consumed more due to feeling depressed or dissatisfied. This approach was adapted from a prior Norwegian survey [37]. EE was defined as a range between 0 (no symptoms in the last week) to 1 (every day). The International Physical Activity Questionnaire—Short Form (IPAQ-SF) [38] was utilized to obtain data on physical activity levels. The IPAQ-SF questions allowed for the measurement of the participants’ total weekly activity energy expenditure (the sum of walking, moderate-intensity physical activities, and vigorous-intensity physical activities) in metabolic equivalent task minutes per week (MET—min/week). High physical activity is defined by the IPAQ-SF scoring rules (http://www.ipaq.ki.se) as >1 h of moderate-intensity activity above basal levels or >30 min of vigorous-intensity activity above basal levels each day. Moderate activity is defined as at least 30 min of moderate-intensity activity on most weekdays. Low activity describes those subjects that do not match the two requirements detailed above. Participants were categorized into the following three physical activity levels: low, moderate, and high. BMI is internationally regarded as a standard for determining weight load, and BMI (kg/m^2^) = weight (in kg)/height^2^ (in m). Underweight was defined as a BMI < 18.5 kg/m^2^, normal weight as ≥18.5 and <25 kg/m^2^, overweight as ≥25 kg/m^2^, and obesity as ≥30 kg/m^2^ [39].

### 2.3. Data Analyses

Stata SE 16 (StataCorp, College Station, TX, USA) was used to conduct all statistical analyses. Except when otherwise noted, the cut-off for statistical significance was set at *p* < 0.05. The first time point (t_0_), when participants initially filled out the survey, was defined as the baseline. In all analyses, time is reported as years from baseline. The following variables in the model were handled categorically with the reference levels: male, the youngest age group (18–30), lower to no worries, and low physical activity. Psychological distress, EE, and intake of high-sugar food and drinks were continuous, meaning the extreme levels were compared to the lowest levels. Linear mixed-model analyses were used to investigate whether emotional eating, intake of high-sugar food and drinks as an indicator of unhealthy food choices, physical activity, psychological distress, worries, and sociodemographic factors were associated with BMI at baseline and to what extent they were associated with any changes in BMI over time. Most participants did not report considerable changes in psychological distress, worries, or activity levels over time (Appendix A). Consequently, these exposure variables were treated as time-invariant factors and kept constant at the baseline values when analyzing the levels of and changes in outcome variables. Although EE and consumption of high-sugar food and drinks decreased over time, the reductions were to neighboring categories and there were very few changes from extreme to low or the other way (Appendix A). Thus, we used the baseline levels as time-invariant factors in the model. We formulated the linear mixed models as a regression model with a random intercept and fixed slope using maximum likelihood estimation. Interactions between these variables and time were included in the model to determine the time trends if the predictors were associated with changes in BMI. To reduce the risk of selection bias, sensitivity analyses using inverse-probability weighting were conducted. Sankey plots were constructed with Sankeymatic (sankeymatic.com) based on cross tables.

To assess the direct and indirect effects of psychological distress on BMI, we performed structural equation modeling where psychological distress could have an indirect effect on BMI through emotional eating, intake of high-sugar food and drinks, and physical activity, and it could have a direct effect as well. In this model, we used the change in BMI between wave 0 and wave 2 (BMI_t2_-BMI_t0_). Since the BMI change was not distributed normally, we applied a Monte Carlo approach using the Stata “medsem” package to investigate the mediation effect [40]. Based on Baron and Kenny’s approach [41], Sobel’s test was used to investigate the significance of the mediation effect [42]. The same procedure was used to assess the direct and indirect effects of health-related and financial-related worries on BMI change.

### 2.4. Ethics Approval and Consent to Participate

The Regional Ethical Committee for Medical Research for Western Norway (REK 2020/131560) gave its approval to the project. Before responding to the electronic survey, each participant was required to give their informed consent. Confidentiality and the ability to drop out of the research were also guaranteed. The study followed the Declaration of Helsinki’s guidelines for ethical research.

## 3. Results

Among the participants, 64% were female, half were younger than 50 years old, 65% had a university or college education, and 68% were employed before the pandemic. Worries related to health were substantial in 45% of the participants, while around 19% had substantial financial worries and 20% had high levels of psychological distress (Table 1).

At baseline, the mean BMI was 25.7 and this increased to 26.2 in two years (Appendix A). The percentage categorized as obese increased from 14% at baseline to 16% at t_2_ (Appendix A). Changes in BMI categories over time are presented in Figure 2, which includes the participants that provided their weight in the surveys at baseline (t_0_), the 1-year follow-up (t_1_), and the 2-year follow-up (t_2_) (*n* = 8218). At baseline, around half of the participants had normal BMI, while one tenth among those were later categorized as overweight. More than one third of the participants were overweight at the one-year follow-up, and among those, one tenth were later categorized as obese at the third time point. Despite some fluctuations between categories through time points, the overall trend was that BMI increased over time towards overweight and obese categories.

Results from the linear mixed model (Table 2) showed an overall time trend of around 0.15 (95% CI: 0.08; 0.23) per year. Comparisons between age groups showed that individuals aged 40–60 years had the highest BMI at baseline (1.53 (95% CL 1.40; 1.66)), but the age difference became slightly less pronounced over time (−0.10 (95% CL −0.15; −0.05)). Female sex compared to male was associated with lower BMI at baseline (−1.47 (95% CL −1.58; −1.36)), but this difference was reduced over time (0.07 (95% CL 0.03; 0.10). Extreme levels of psychological distress compared with no psychological distress were associated with a higher BMI at baseline 1.62 (95% CI 1.27; 1.97). The time trend for this association was positive (0.26 (95% CL 0.10, 036)), meaning those who had extreme psychological distress also gained more weight over time compared to those who had no distress. EE episodes every day compared with no EE were positively associated with higher BMI at baseline 0.68 (95% CI 0.57; 0.72). The time trend for this association was also substantially positive, showing that having episodes of EE every day was associated with increases in BMI by 0.27 (95% CI 0.17; 0.37) per year compared to those without EE. Daily intake of high-sugar food and drink compared to rare/no intake was also associated with higher BMI at baseline (0.35, 95% CI 0.20; 0.49). Moderate and high physical activity compared to low activity were associated with −0.16 (95% CL −0.23; −0.10) and −0.27 (95% CL −0.34; −0.20) lower BMI at baseline; however, these associations were reduced over time (−0.01 (95% CL −0.06; 0.04)). Sensitivity analyses of the linear mixed models using inverse-probability weights provided similar results as the standard models (Appendix A).

The mediation effect of EE on BMI change was partial (Table 3), with 27% of the effects of psychological distress on BMI change being mediated by EE (indirect effect/total effect = 0.016/0.059). There were no mediation effects of the intake of high-sugar foods and drinks or physical activity on BMI change in this model. Furthermore, we investigated the direct and indirect effects of health-related worries on BMI change (Appendix A). There was a partial mediation effect of EE on BMI change, meaning that 18% of the effects of health-related worries on BMI change were mediated by EE (indirect effect/total effect = 0.008/0.043). Additionally, results from the analysis of direct and indirect effects of financial worries on BMI change (Appendix A) showed a complete mediation effect for EE, meaning that all the effects of financial worries on BMI change were mediated by EE in this model.

## 4. Discussion

In this study, we revealed that the mean BMI increased from 25.7 to 26.2 during the two first years after the initiation of the COVID-19 pandemic, with increased proportions of overweight and obese. High psychological distress was strongly associated with higher BMI in the early phase of the pandemic and was also strongly associated with an increase in BMI over time. Moderate and high activity were associated with lower BMI at baseline compared to low activity, with a tendency toward a similar change over time. Emotional eating, based on our short questionnaire tool, was associated with higher weight during the early phase of the pandemic and with a substantial increase in BMI over time. Additionally, the effects of psychological distress and health-related worries on change in BMI were partially mediated by EE, while the effects of financial worries on BMI were completely mediated by EE. Males had generally higher BMI levels than females. Similarly, older adults had higher BMI than younger adults, but the differences between these reduced over time.

Our findings are consistent with previous research reporting that emotional eating is related to weight gain [43,44,45] but add some precision, particularly for settings beyond the United States. Certain intense emotions, such as anxiety, restlessness, anger, fear, happiness, and sadness, can contribute to major changes in eating behaviors by boosting the desire to eat, the volume of food ingested, and the selection of unhealthy meals [16]. Weight can be influenced by any of these eating behaviors, so simply avoiding unhealthy food items like high-sugar products while eating more in response to emotions may not be sufficient to prevent weight gain. What is seen as “comfort food” may also differ between individuals and is not necessarily restricted to high-sugar foods and drinks [46]. This may explain our finding that intake of high-sugar food and drinks was not significantly associated with higher BMI over time.

It is known that moderate regular physical activity contributes to better health in various ways [47,48]. A study in Brazil during the COVID-19 pandemic showed that those who were physically active (>150 min/week) reported less EE. They had lower levels of stress and lower consumption of sweet food items [49]. This is in line with our results for the association between moderate and high activity and lower BMI.

At baseline, the daily activity of the participants was in general severely limited due to the preventive restrictions of the pandemic, and this could have contributed to the initial weight gain during the first year of the pandemic. One study assessing the longitudinal weight change in US adults during the pandemic (peak-lockdown compared to post-lockdown era) reported an increase in average BMI, as well as sedentary behaviors, in those who gained weight [25]. Another study in France reported decreased physical activity in 53% and increased sedentary behaviors in 64% of the participants during lockdown, with 35% of them gaining weight. In our study, even with substantial reductions in restrictions during the second year of the pandemic, we observed an increase in BMI that could be attributed to the participants’ advancing age. Also, it might take some time for the participants to revert to their pre-pandemic lifestyle.

Our study showed that high psychological distress was strongly associated with higher BMI at baseline, with a stronger association over time. The effects of psychological distress on BMI were mediated partially by EE. This suggests that factors other than EE may contribute to obesity. In our previous study on the same cohort, we established that EE was reduced over time during different phases of the pandemic [50]. We argued that one possible explanation could be related to the state of chronic stress. In this state, weight gain and subsequent additional abdominal fat tissue could trigger a negative feedback mechanism, leading to the suppression of hormone release, such as cortisol, which is commonly associated with stress [51]. In other words, those who experience high distress might gain weight due to stress-related hormones and also due to changes in their behavior during a stressful situation.

A recent study assessed the associations between emotional dysregulation, psychological distress, EE, and BMI in a cross-sectional setting and reported that emotional dysregulation was linked to higher psychological distress and EE, which was associated with a higher BMI [52]. Our findings also underscore the significance of using emotional regulation techniques to address EE in stressful situations [53] rather than solely relying on dietary advice. For instance, during events like the COVID-19 pandemic, guidelines issued by authorities that promote healthy food choices and physical activity may be less effective due to individuals’ lack of impulse control. As a result, prioritizing immediate relief rather than long-term health goals can undermine the ability to adhere to these guidelines [54]. Mindfulness training has been shown to decrease EE in affected populations [55] and could be used along with other public prevention measures in future similar stressful situations.

One strength of this study was the large sample size, which implied that estimates had improved precision and statistical power compared to most studies. The annual follow-ups of the individuals gave perspective into changes that occurred throughout the initial and later periods of the pandemic. The cohort is likely generalizable to some extent to other groups in high-income countries. In addition, our sample consisted of both male and female respondents with a large age span. The findings must be considered in the context of the study’s limitations. First, self-reporting and reliance on the participants’ impressions made our findings prone to recall bias. However, because the memory period was quite brief, recall bias was expected to be less relevant. Second, the questionnaire was based on validated questions concerning psychological distress and worries, but there were relatively few questions that have been used in other large population studies to address eating habits, which would have allowed for comparisons. It may have contained less information and fewer nuances than a longer, more quantitative questionnaire, but longer surveys risk losing participants due to questionnaire fatigue. Thirdly, our study had a small inherent selection bias due to the questionnaire being in Norwegian and digital; therefore, people with limited Web access or who had limited knowledge of or were not fluent in the Norwegian language could not participate. Thus, some of the elderly and first-generation immigrants might have been underrepresented, but the elderly had a higher response rate when approached, which balanced some of the potential imbalance. Furthermore, since the results of inverse-probability weighting were similar to the standard model, an effect from selection bias on the results was unlikely. Fourth, loss to follow-up of approximately half of the participants through the two-year period could have contributed to increasing selection biases. Nevertheless, the background characteristics in each group were comparable; thus, we think the observed relationships were probably not substantially affected by selection bias [56].

## 5. Conclusions

Our results suggest that the mean weight of the population increased during the two first years after the COVID-19 pandemic struck. Increased body mass index was strongly associated with psychological distress, emotional eating, and physical inactivity. Furthermore, the effects of psychological distress on body weight were partially mediated by emotional eating. To modify a trend of increasing obesity, interventions aiming to reduce psychological distress and thus also emotional eating could be considered. Follow-up studies assessing BMI after the termination of pandemic restrictions are needed to prevent health-related consequences of increasing BMI in the general population.

## Figures and Tables

**Figure 1 nutrients-15-03834-f001:**
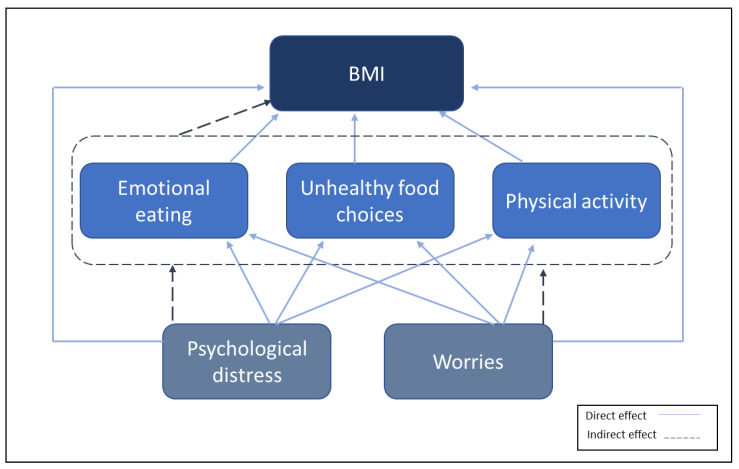
Hypothesis model of the association between variables.

**Figure 2 nutrients-15-03834-f002:**
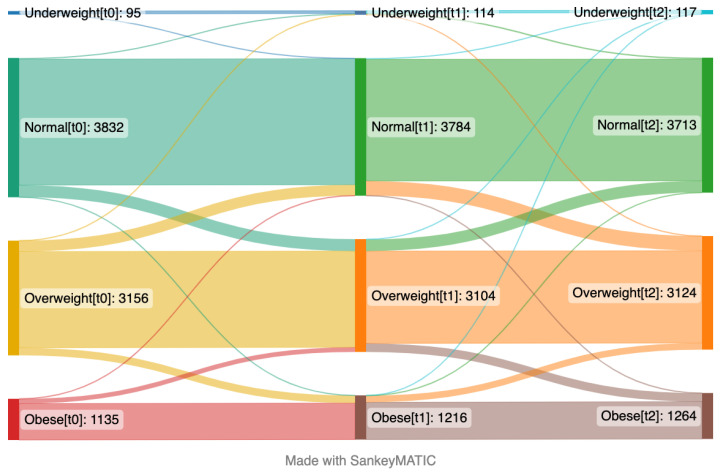
Sankey plot presenting proportions of participants’ changes between different body mass index categories over time from April 2020 (left side, t_0_) to January 2021 (middle, t_1_) and May 2022 (right side, t_2_). The numbers in brackets represent time points. This plot includes the participants who provided their weight and height in the questionnaires at the first time point (t_0_), second time point (t_1_), and third time point (t_2_) (*n* = 8218).

**Table 1 nutrients-15-03834-t001:** Background characteristics of the participants (%).

Age	18–40	40–60	60+	Total
Age distribution (%)	7131 (31)	9328 (41)	6489 (28)	22,948
Sex (%)				
Female	4385 (61)	5270 (56)	3017 (46)	12,672 (55)
Male	2746 (39)	4058 (44)	3472 (54)	10,276 (45)
BMI median (25–75%)	24 (22–27)	26 (23–28)	25 (23–28)	25 (23–28)
BMI categories (%)				
Underweight	172 (2)	65 (1)	65 (1)	302 (1)
Normal	4135 (58)	4031 (43)	2883 (44)	11,049 (48)
Overweight	2012 (28)	3738 (40)	2736 (42)	8486 (37)
Obese	812 (11)	1494 (16)	805 (12)	3111 (14)
Education level *				
Primary school	533 (7)	430 (5)	610 (9)	1573 (7)
High school/trade school	1868 (26)	2350 (25)	2039 (39)	6257 (27)
≤3 years of higher education	1862 (26)	2249 (24)	1494 (23)	5606 (24)
≥4 years of higher education	2864 (40)	4287 (46)	2324 (36)	9475 (41)
Employment prior to COVID-19 (%) **				
Employed (full/part-time)	5555 (78)	8354 (90)	2458 (38)	16,367 (71)
Student	1779 (25)	126 (1)	9 (0)	1914 (8)
COVID-19 consequences in the initial period (%) **				
Temporarily laid-off	866 (12)	731 (8)	222 (3)	1819 (8)
Lost employment	132 (2)	74 (1)	26 (0)	232 (1)
Home office	1358 (19)	1373 (15)	1013 (16)	3744 (16)
Placed in quarantine	2712 (38)	4877 (52)	1248 (19)	8837 (39)
Substantial worries (%)				
Related to personal economy	1875 (26)	1600 (17)	368 (6)	3843 (17)
Health-related	3698 (52)	4333 (46)	2137 (33)	10,168 (44)
High psychological distress (%)	2419 (34)	1593 (17)	619 (10)	4631 (20)

* Total number of participants who provided their educational level = 22,910. ** Alternatives were checkboxes that were not mutually exclusive.

**Table 2 nutrients-15-03834-t002:** BMI and its associations with exposure variables at baseline and over time are presented with coefficients (with 95% confidence intervals) *.

	Fixed Effects	Time Trend (Per Year)
Time trends per year **		0.15 (0.08; 0.23)
Age		
18–40	0 (reference)	0 (reference)
40–60	1.53 (1.40; 1.66)	−0.10 (−0.15; −0.05)
60+	1.17 (1.02; 1.32)	−0.24 (−0.29; −0.19)
Sex		
Male	0 (reference)	0 (reference)
Female	−1.47 (−1.58; −1.36)	0.07 (0.03; 0.10)
Health-related worries		
None or some	0 (reference)	0 (reference)
Substantial	−0.02 (−0.08; 0.03)	0.05 (0.01; 0.10)
Worries related to economy		
None or some	0 (reference)	0 (reference)
Substantial	0.05 (−0.02; 0.14)	−0.02 (−0.09; 0.05)
Psychological distress (0 = none to 1 = extreme)	1.62 (1.27; 1.97)	0.23 (0.10; 0.36)
Emotional eating (0 = never to 1 = everyday)	0.68 (0.57; 0.79)	0.27 (0.17; 0.37)
High-sugar foods and drink intake (0 = no to 1 = daily)	0.35 (0.20; 0.49)	0.10 (0.00; 0.20)
Activity level		
Low	0 (reference)	0 (reference)
Moderate	−0.16 (−0.23; −0.10)	−0.01 (−0.06; 0.04)
High	−0.27 (−0.34; −0.20)	−0.05 (−0.11; 0,00)

* Linear mixed model presenting absolute coefficients, with 0 indicating no difference/change. ** Adjusted baseline coefficient of BMI in this model: 25.22 (25.06; 25.37).

**Table 3 nutrients-15-03834-t003:** Mediation of associations between psychological distress and BMI presented with coefficients (with 95% confidence intervals).

Variable	Direct Effect	Indirect Effect	Total Effect	β
BMI difference				
Psychological distress	0.17 (0.08; 0.28)	0.06 (0.02; 0.11)	0.24 (0.15; 0.33)	0.04 *
Emotional eating	0.04 (0.01; 0.07)		0.04 (0.01; 0.07)	0.03 *
High-sugar foods and drink intake	−0.01 (−0.03; 0.02)		−0.01 (−0.03; 0.02)	−0.004
Physical activity	−0.01 (−0.07; 0.05)		−0.01 (−0.07; 0.05)	−0.004
Emotional eating				
Psychological distress	1.55 (1.49; 1.62)		1.55 (1.49; 1.62)	0.47 *
High-sugar foods and drink intake				
Psychological distress	0.49 (0.41; 0.56)		0.49 (0.41; 0.56)	0.14 *
Physical activity				
Psychological distress	−0.18 (−0.21; −0.14)		−0.18 (−0.21; −0.14)	−0.12 *

* *p* value < 0.05.

## Data Availability

The data presented in this study are available on request from the corresponding author.

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
