# Peer review of "Changes in Body Mass Index and Their Associations with Psychological Distress, Worries, and Emotional Eating during the COVID-19 Pandemic: A Norwegian Cohort Study"

_nutrients, 2023, doi:10.3390/nu15173834_

Round 1

Reviewer 1 Report

This is an interesting, well written and large study, dealing with a very interesting topic.

Nevertheless I have some concerns, major and minor.

row 48 "excreta" is wrong: the english term is "excetera", the correct latin term "etcetera" or "et cetera";

row 167 the abbreviation SEM is potentially confusing, since it is the globally accepted short of "standard error of the mean", so it should be accordingly changed;

Table 1: the figures don't add up in: educational level, employment, and Covid consequences;

Figure 2 seems cumbersome and very difficult to understand;

row 295 "high precision" seems exaggerated and should be substituted with "major precision" or "improved precision";

row  316 the authors claim "the mean weight increased substantially during...", while the increase of BMI is a mere 2%. The authors should review the tone of their conclusions.

References should be reviewed, too. A simple search on Pubmed/Medline retrieved some 175 papers dealing with the same topics of the present study, from 2020 to 2023. I suggest to update accordingly the reference section

Reviewer 2 Report

This is appropriate and interesting topic and address the thematic of the Journal. The article provides helpful insight related to relations between changes in body mass index and psychological distress and emotional eating but some revisions are needed to be clearer and to improve the quality of the article.

Overall the literature review and purpose of article was well-developed taking into account the impacts of the pandemic that increased psychological distress, emotional eating and a more sedentary life styles, but it would be helpful for the authors to read through again with the intention of being even more precise and documenting more specifically the psychological distress term since it is highlight in the title and also through the article and also related to the term of emotional eating and worries.

Methods:  The research design, questions, hypotheses and methods are clearly stated.

Although some questions raised:

Regarding the sample, any specific exclusion criteria?  

Regarding the instrument:

-I think that the questionnaire focuses mainly in psychological distress and worries, and not in eating habits, that is also a focus of the research and paper. It would be helpful to address and explain better this and how this would have or not allowed for comparison (related to the main aim of the study);

 - I suggest to explain better how the authors overcome the constraint of a digital format of the instrument and also language barriers, because this can compromise the study

Results and discussions: The results are organized and clearly stated. But I think that considering your sample and results would be interesting to address in the discussion sections some aspects:

Do you have specific consideration about gender issues? And also related to ages?

Conclusion: I did feel a need for a deeper conclusion at the end of article that summarizes the principles themes/objectives that appeared in the introduction section - weaving back to recent highlights in the literature.

Good organization of the paper and references: Another read through to check minor grammar errors and references norms would be helpful

Another read through to check minor grammar errors and references norms would be helpful.

Round 2

Reviewer 1 Report

I appreciated the effort to improve the paper accordingly to the suggestions I presented. In my opinion the paper seems acceptable for publication now.